# Fast photostable expansion microscopy using QDots and deconvolution

**Loku Gunawardhana**[1], **Wilna Moree**[2], **Jiaming Guo**[1], **Camille Artur**[1], **Tasha Womack**[3], **Jason L. Eriksen**[3], **David Mayerich** [1]*

**1** University of Houston, Department of Electrical and Computer Engineering, Houston, Texas, United States of America, **2** SwiftFront, LLC, Houston, Texas, United States of America, **3** University of Houston, Department of Pharmacological and Pharmaceutical Sciences, Houston, Texas, United States of America

* mayerich@uh.edu

**Data availability statement:** Raw data used for plots of time series (Fig 2) or point spread

## Abstract

Expansion microscopy (ExM) enables sub-diffraction imaging by physically expanding labeled tissue samples. This increases the tissue volume relative to the instrument point spread function (PSF), thereby improving the effective resolution by reported factors of 4 - 20X. However, this volume increase dilutes the fluorescence signal, reducing both signal-to-noise ratio (SNR) and acquisition speed. This paper proposes and validates a method for mitigating these challenges. We overcame the limitations of ExM by developing a fast photo-stable protocol to enable scalable widefield three-dimensional imaging with ExM. We combined widefield imaging with quantum dots (QDots). Widefield imaging provides a significantly faster acquisition of a single field-of-view (FOV). However, the uncontrolled incoherent illumination induces photobleaching. We mitigated this challenge using QDots, which exhibit a long fluorescence lifetime and improved photostability. First, we developed a protocol for QDot labeling. Next, we utilized widefield imaging to obtain 3D image stacks and applied deconvolution, which is feasible due to reduced scattering in ExM samples. We show that increased transparency, which is a side-effect of ExM, enables widefield deconvolution, dramatically reducing the acquisition time for three-dimensional images compared to laser scanning microscopy. The proposed QDot labeling protocol is compatible with ExM and provides enhanced photostability compared to traditional fluorescent dyes. Widefield imaging significantly improves SNR and acquisition speed compared to conventional confocal microscopy. Combining widefield imaging with QDot labeling and deconvolution has the potential to be applied to ExM for faster imaging of large three-dimensional samples with improved SNR.

## 1 Introduction

Expansion microscopy (ExM) [1–4] is an inexpensive and powerful tool for studying super-resolved tissue microstructures using conventional epifluorescence microscopes. The sample is labeled and embedded within a swellable hydrogel. Proteins are then digested, and the embedded labels physically expanded by adding water. ExM provides locally isotropic expansion with minimal distortion, improving both lateral and axial resolution by several factors

functions (Fig 4) are provided as supplementary data in Excel files. 3D images are several gigabytes in size, making them incompatible with any resource sharing sites that provide a DOI. However, example cross-sections for PSF were provided as TIF files.

**Funding:** (DM) National Institutes of Health/National Heart, Lung, and Blood Institute (NHLBI) #R01HL146745 (https://www.nhlbi. nih.gov/). (DM) National Science Foundation CAREER Award #1943455 (https://www.nsf.gov/). Sponsors did not play a role in the study design, data collection, analysis, decision to publish, or preparation of the manuscript.

**Competing interests:** Jason Eriksen and David Mayerich have an ownership stake in SwiftFront, LLC. Wilna Moree is an employee of SwiftFront, LLC and was provided salary support. This does not alter our adherence to PLOS ONE policies on sharing data and materials.

(approximately 4X) [1] with iterative techniques reporting up to 20X [4]. The digestion process breaks down proteins and removes lipids, reducing scattering similar to optical clearing methods [5–8].

ExM primarily relies on confocal imaging to acquire 3D images using fluorescent dyes [1–7,9]. The decrease in fluorophore density, combined with digestion-induced quenching [10], decreases signal-to-noise ratio (SNR) and lengthens acquisition time. Widefield imaging can compensate by using a broadband incoherent light source [2,5,11]. However, this increases illumination incident on the sample, causing photobleaching while reducing z-axis resolution. We demonstrate that these limitations can be improved by combining quantum dots (QDots) with widefield deconvolution, while taking advantage of the reduced scattering that occurs with ExM. This combined approach makes widefield imaging practical, since QDots minimize photobleaching, while deconvolution improves axial resolution.

QDots exhibit a long fluorescence lifetime and resistance to photobleaching, allowing image acquisition with minimal signal loss. Their improved photostability and higher quantum yield provide advantages when the number of fluorophores is low. In addition, their narrow emission bandwidth and deep UV excitability make them useful for multiplexing applications [12,13]. However, QDots tend to aggregate in buffer, owing to their non-optimal surface chemistry and larger size, which may hinder access to target biomolecules. Moreover, there is no consensus method for labeling, and only a few standard protocols are available [14–17]. Previously reported expansion nanoscopic imaging (ExN) utilized QDots for labeling clathrin-coated pits and microtubules in cultured cells [18]. However, due to their poor penetration ability, there are currently no published protocols supporting QDot labeling of tissue with ExM. Therefore, QDots have not been extensively used in ExM immunostaining to date.

By combining the improved photostability and quantum yield of QDots [19–21] with ExM, we demonstrate three-dimensional widefield acquisition of super-resolved tissue using multiple fluorescent channels. These protocols provide an additional tool to significantly increase the speed of ExM imaging without laser-scanning confocal or multi-photon microscopes.

## 2 Materials and methods

### 2.1 Reagents and antibodies

Primary antibodies for immunostaining were: anti-GFAP (ab7260, Abcam), anti-MBP (MAB386, Millipore Sigma), and anti-$\alpha$-tubulin (T8203, Millipore Sigma). Biotinylated antibodies were used for secondary labeling: goat anti-rabbit (111-065-144, Jackson immuno), goat anti-mouse (115-065-146, Jackson immuno), and goat anti-rat (A24553, Invitrogen). Normal goat serum (NGS, Sigma-Aldrich) was used for antibody labeling. Xylene substitute was purchased from Sigma for tissue deparaffinization. Streptavidin functionalized Qdot 585 (Q10111MP), Qdot 655 (Q10123MP), and Alexa Fluor 594 (S32356), as well as Hoechst 34580 (H21486) were obtained from Thermo fisher scientific. ExM reagents include: (6-((acryloyl)amino)hexanoic acid, succinimidyl ester) (Acryloyl-X, SE, AcX, Fisher), dimethyl sulfoxide (DMSO, Sigma), sodium acrylate (Sigma), acrylamide (Sigma), N,N′-methylbisacrylamide (Sigma), sodium chloride (Fisher), 4-hydroxy-TEMPO (Sigma), tetramethylethylenediamine (TEMED, Sigma), ammonium persulfate (APS, Sigma), Proteinase K (New England Biolabs), tris pH 8.0 (Sigma), EDTA (Sigma), triton X-100 (Sigma), guanidine HCl (Sigma), and Fluoresbrite YG 1.0 $\mu$m microspheres (polysciences).

## 2.2 Immunostaining protocol

Formalin free fixed (Accustain) paraffinized mouse brain was cut into 10 $\mu$m sections and placed on superfrost plus glass slides. Tissues were deparaffinized using xylene substitute and re-hydrated in a graded series of ethanol. Glass-mounted sections were washed with 0.1% Tween 20 in 1X Tris-buffered saline (TBST), and an additional permeabilization step to guide QDots into cells was carried out by incubating the tissues with permeabilizing reagent (0.2% Triton X-100 in PBS) for 10 minutes two times, consecutively. Slices were washed with TBST and incubated with a blocking buffer consisting of 5% normal goat serum (NGS) in TBST for 1 hour followed by 1 hour incubation of primary antibodies in 5% NGS in TBST at room temperature. Primary antibodies used in the study were: anti-GFAP (1:250), anti-MBP (1:250) and anti-$\alpha$-tubulin (1:250). Sections were incubated with biotinylated goat anti-rabbit antibody in TBST (1:200) for 1 hour at room temperature followed by the incubation of streptavidin conjugated Qdot 585 (1:25) in TBST for 1 hour at room temperature. Sections were blocked for avidin/biotin (Vector) as per the manufacturer's instructions, before treating the tissue with the next secondary antibody to prevent cross reactivity and high background fluorescence. Sections were then incubated with biotinylated goat anti-rat antibody or biotinylated goat anti-mouse antibody in TBST (1:200) for 1 hour at room temperature followed by incubation with streptavidin conjugated Qdot 655 (1:25) in TBST for 1 hour at room temperature. Finally, the tissues were co-stained with Hoechst 34580 at a 1:1000 dilution in TBST for 30 minutes. Between each incubation, sections were washed three times with TBST for 10 minutes each.

## 2.3 Gelation, digestion, and expansion of mouse brain tissue

Brain tissues were expanded following a previously published protocol [5]. Immunostained sections were incubated with AcX in 1x PBS at a dilution of 1:100 overnight at room temperature. The AcX solution was made by dissolving AcX in DMSO at a concentration of 0.1 mg/mL. Prior to the next step, the monomer solution was prepared by mixing 2.25 mL of 0.38 g/mL sodium acrylate, 0.5 mL of 0.5 g/mL acrylamide, 0.75 mL of 0.02 g/mL N,N′-methylbisacrylamide, 4 mL of 0.29 g/mL sodium chloride, 1 mL PBS 10X and 0.9 mL deionized water. The gelling solution was prepared by mixing 188 $\mu$l of monomer solution, 4 $\mu$l of 0.5% 4-hydroxy-TEMPO, 4 $\mu$l of 10% TEMED, and 4 $\mu$l of 10% APS. Sections were then rinsed with PBS twice for 15 minutes each and incubated with gelling solution for 30 minutes at 4°C. After 30 minutes, the sections were transferred to a gel chamber made with a No. 1 coverslip and incubated with freshly prepared gelling solution for 2 hours at 37°C. After gelation, the excess gel was trimmed off and sections were digested with proteinase K 1:100 in the digestion buffer (50 mM tris pH 8.0, 1 mM EDTA, 0.5% triton X-100, 0.8 M guanidine HCl) overnight at room temperature. Digested gels were rinsed with 1x PBS for 15 minutes and then washed in deionized water multiple times to expand until the expansion reached a plateau.

## 2.4 Point spread function measurement

Mouse brain was incubated in a cryo-protectant solution overnight and then covered with cryo-embedding media (OCT) and stored at –80 °C for 30 minutes. Frozen blocks were transferred to a cryotome cryostat (Leica CM1850) and cut into (10 $\mu$m) sections. The tissue sections were washed with PBS 1x for 5 minutes and placed on a glass slide. A drop of fluorescent yellow micro-beads (1 $\mu$m) in a solution of 1x PBS was added to cover the tissue section, which was then folded using a paintbrush (Fig 4a). Folded sections were washed with 1x PBS

to remove extra beads. These folded sections were imaged both pre- and post-expansion (see protocol below) to provide a comparison between point spread functions (PSFs).

## 2.5 Imaging and deconvolution

Tissue sections were imaged using a commercial wide-field fluorescence microscope (Nikon Eclipse TI-E Inverted Microscope). Qdot 585, and Qdot 655 labeled samples were excited at the wavelength of 488 nm. Hoechst 34580 labeled samples were excited at the wavelength of 365 nm, and Alexa 594 labeled samples were excited at the wavelength of 561 nm. All images were collected using a 40X water-immersion objective (Nikon N40X-NIR, 0.8NA, 3.5 mm WD), with a lateral sampling rate of 0.16 μm per pixel, coupling the refractive index of the embedding gel to minimize longitudinal elongation. Two focal stacks (pre- and post-expansion) were collected at an interval of 0.8 μm, in which the expanded gel was immobilized inside a glass-bottom petri-dish for post-expansion imaging. Excess water was removed to mitigate tissue drift during imaging.

Raw z-stack images were deconvolved using Huygens Software (Scientific Volume Imaging) with classic maximum likelihood estimation (CMLE) based on a theoretical PSF using known microscope parameters (above). All processed images were visualized and registered using ImageJ.

For comparison, the expanded tissue was imaged with a confocal system (Leica SP8 3X). A 3D image stack was acquired at the same thickness with a 40X objective (0.6NA), and the results were compared with the wide-field stack acquired with a 40X air-immersion objective (0.6NA).

# 3 Results and discussion

We demonstrate the combined QDot/ExM approach on mouse brain tissue using immunofluorescent markers for glial fibrillary acidic protein (GFAP), myelin basic protein (MBP), and alpha-tubulin. Hoechst 34580 was used as a counterstain to identify cell nuclei. GFAP is the most popular marker for astrocytes in neurological studies, enabling visualization of the cell body and processes. It is also an intermediate filament protein released after cell death or injury used as a marker for the diagnosis of brain injury [22,23]. MBP is a trans-membrane protein in the nervous system that maintains myelin structure by interacting with lipids in the myelin membrane [24,25]. Alpha-tubulin is a cytoskeleton protein which is a building block of microtubule filaments [26], and is commonly used in super-resolution experiments due to its fine detail and interesting filamentary structure.

## 3.1 Immunostaining with QDots

QDot 585 and QDot 655 staining were validated for immunostaining in mouse brain tissue before performing the ExM protocol. Both QDots exhibit broad excitation bandwidths in the lower visible range and narrow emission bandwidths in the higher visible and near-infrared ranges, making them ideal for multiplexed fluorescence mapping. Three common biomarkers were observed using a single (488 nm) excitation wavelength: GFAP-positive astrocytes were labeled with QDot 585, MBP-positive myelin was labeled with QDot 655, and tubulin-positive microtubules were labeled with QDot 655 in separate mouse brain tissues. These QDot stainings were in accordance with the fluorescence staining using Alexa Fluor (Thermo Fisher Scientific) dyes.

### 3.2 Protein-specific triple-staining

Before carrying out the expansion protocol, we performed triple-staining on mouse brain tissues to confirm the potential for QDot multiplex labeling as a pre-expansion assessment. Mouse hippocampus was labeled with GFAP (Qdot 585) (Fig 1a), MBP (Qdot 655) (Fig 1b), Hoechst 34580 (Fig 1c) as a nuclear counterstain, and an overlay of the three stains (Fig 1d). Fig 1e–1h show the magnified views of boxed regions of Fig 1a–1d, respectively for improved resolution. Fig 1d and h show successful triple-labeling in the mouse hippocampus. The resolution of conventional optical microscopes is diffraction-limited, making it difficult to reveal subcellular structures, while widefield ExM is capable of revealing super-resolved cellular structures.

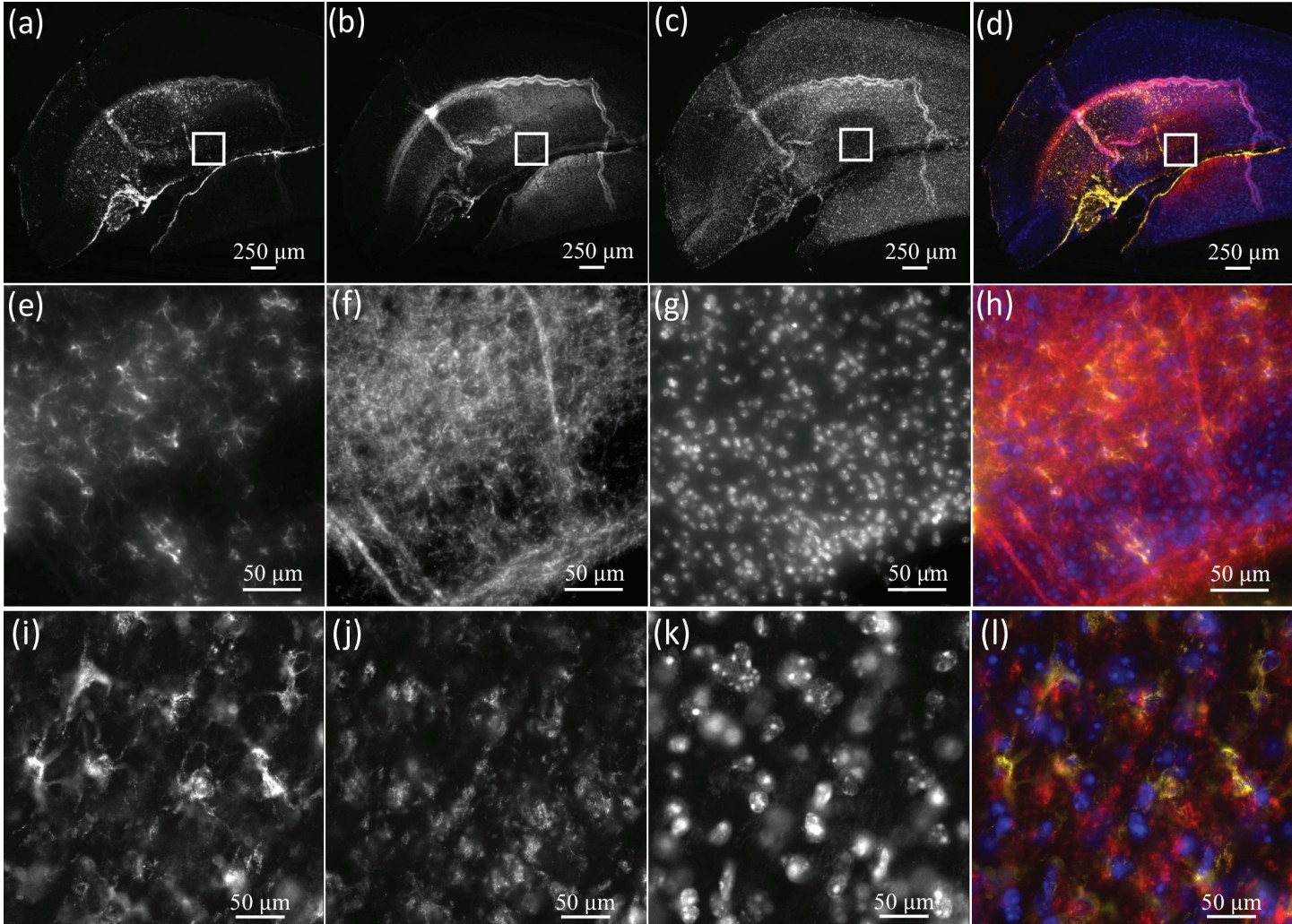

**Fig 1. Pre-expansion (a-h) images of triple-stained mouse brain tissue imaged using a widefield fluorescence microscope.** The tissue was successively stained for GFAP-positive astrocytes (QDot 585, yellow) (a), MBP-positive myelin (QDot 655, red) (b), Hoechst 34580 (blue) (c) and imaged using a 4X (0.13NA) and 40x water dipping objective (0.8NA) (e-g) respectively. Triple stained tissue (d) with 4X (0.13NA) and 40x water dipping objective (0.8NA) (h). Post-expansion (i-l) images of GFAP-positive astrocytes (QDot 585, yellow) (i), MBP-positive myelin (QDot 655, red) (j), Hoechst (blue) (k) and triple imaging (l) using the same 40x objective shows the preservation of all labels. The scale bar represents 50 $\mu$m (physical size) to allow comparison with the pre-expansion image (h). However, expansion microscopy physically magnified the sample $\approx$4.1X, so the 50 $\mu$m scale represents $\approx$ 12.2 $\mu$m structurally.

### 3.3 QDot expansion microscopy

We then applied ExM to the triple-stained mouse brain tissue to prove the stability of QDot staining after expansion. Fig 1 shows single two-dimensional ExM images of GFAP-positive astrocytes (QDot 585, yellow) (Fig 1i), MBP-positive myelin (QDot 655, red) (Fig 1j), Hoechst 34580 (blue) (Fig 1k) and triple-stained mouse brain tissue (Fig 1l). The physical magnification of ExM enables super-resolution imaging on a diffraction-limited microscope with reduced scattering, since the sample is mostly water. We evaluated the expansion factor ($\approx$ 4.1X) with Hoechst-stained nuclei by measuring the longest diameter of the structure before and after expansion. We further confirmed this factor by analyzing pre-and post-expansion sizes of astrocyte cells. These post-ExM images prove the feasibility of utilizing QDots for multiplex ExM imaging.

### 3.4 QDot photostability comparison with conventional dyes

The photostability of QDot was compared with traditional fluorophores using time-lapse measurements of MBP-immunolabeled mouse brain tissues before expansion (Fig 2). Time-lapse experiments revealed that the initial fluorescence intensity of MBP-positive cells labeled with Alexa 594 experienced an 85% drop in fluorescence within 10 minutes, increasing to 98% after 30 minutes of excitation. In contrast, QDot 655 labels are highly stable. No significant intensity loss was observed within the first 50 minutes of continuous exposure. In addition, a slight increase in fluorescence was observed at t = 30 min, possibly due to additional binding. [27,28].

To further confirm the photostability of QDots, we performed time-lapse measurements of Hoechst 34580 labeled nuclei and QDot 585 labeled astrocytes after expansion (Fig 3). Cells labeled using Hoechst 34580 (Fig 3) lose signal completely within the first 5 minutes of exposure while QDots labeled cells show only a 8.7 % intensity loss after 30 minutes of excitation. This exceptional photostability confirms that Qdots are more robust than traditional fluorophores for long-term imaging in medical applications. While the quantum dots used in this study are created through a proprietary manufacturing process, these results are consistent with previous comparisons of QDots and Alexa Fluor dyes [29]. The goal of this demonstration is to validate that QDots are compatible with the aqueous environment required for ExM.

### 3.5 Point spread function characterization

Before performing deconvolution on the expanded labeled tissues for improved resolution, we first assessed the impact of the sample on the stability of the PSF, since the precision of image deconvolution is highly dependent on the consistency of the PSF model.

Biological specimens are thicker (>5 µm) and have a variable refractive index, resulting in large heterogeneous PSFs due to scattering. In comparison, expansion microscopy provides the advantage of reduced light scattering from a more uniform refractive index. We assess the difference between scattering PSFs before and after (Fig 4). Beads closer to the coverslip produce more uniform PSFs compared to beads further into the sample, with uniformity correlated with the amount of tissue penetrated. Average intensity profiles confirm the reduction of light scattering after expansion. The full width at half maximum (FWHM) is used to quantify PSF size. The average FWHM of the radial intensity profile was reduced to 0.584 µm from 0.785 µm after expansion and the axial PSF was reduced to 5.994 µm from 8.755 µm.

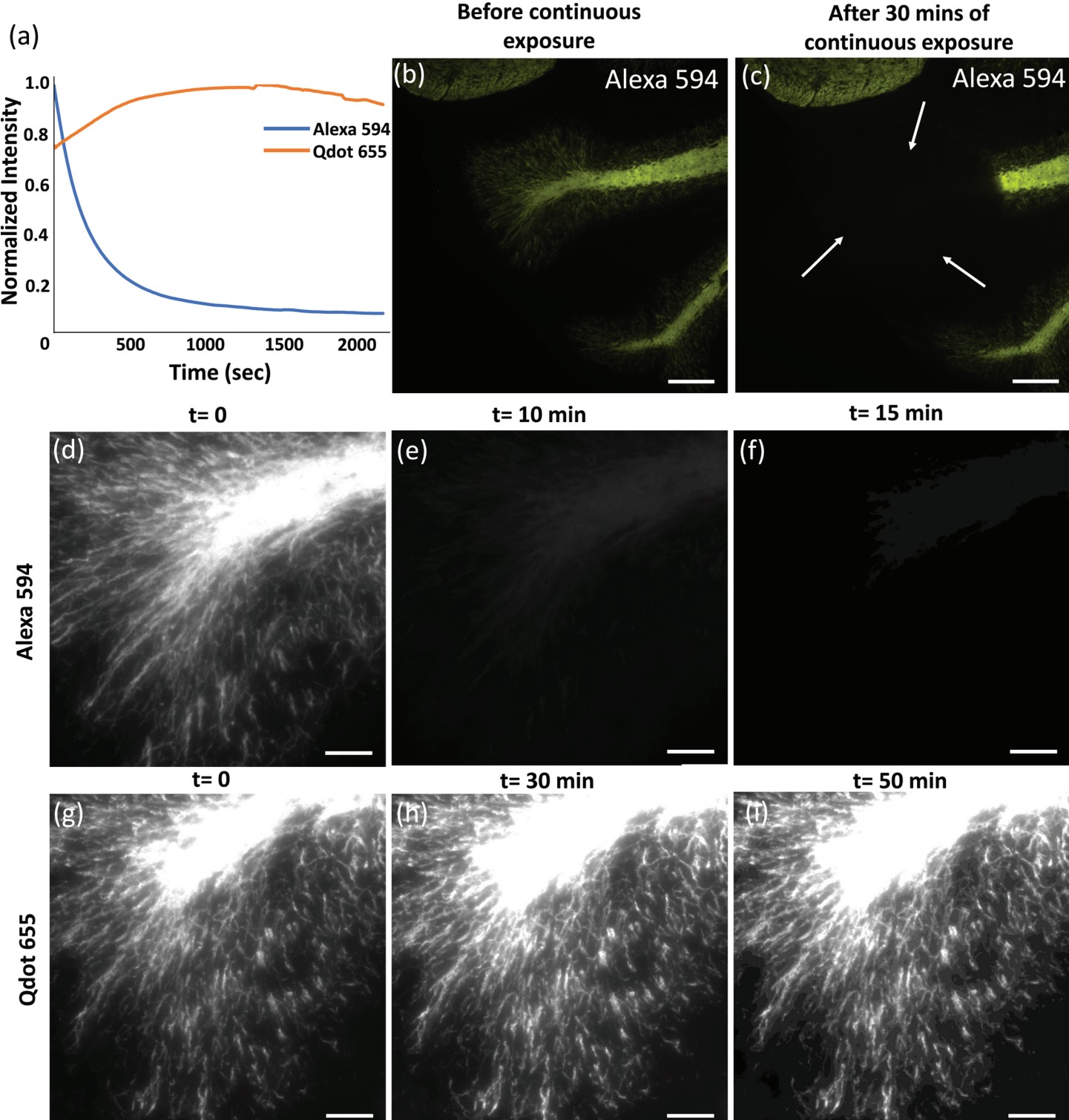

**Fig 2. Photostability comparison of Alexa 594 and QDot 655.** Both labels target MBP. Timelapse measurements were taken for 35 min at 10 s intervals using a 40x water immersion objective (0.8NA). (a) The normalized intensity is plotted over time. Low-magnification images of Alexa 594 samples were obtained at 10X (0.45NA) both before (b) and after (c) 30 minutes of exposure showing the bleached region (arrows) (scale bar 200 $\mu$m). Alexa 594 time points were taken at 0 min (d), 10 min (e), 15 min (f), and QDot 655 time points were taken at 0 min (g), 30 min (h), and 50 min (i) (scale bar 50 $\mu$m).

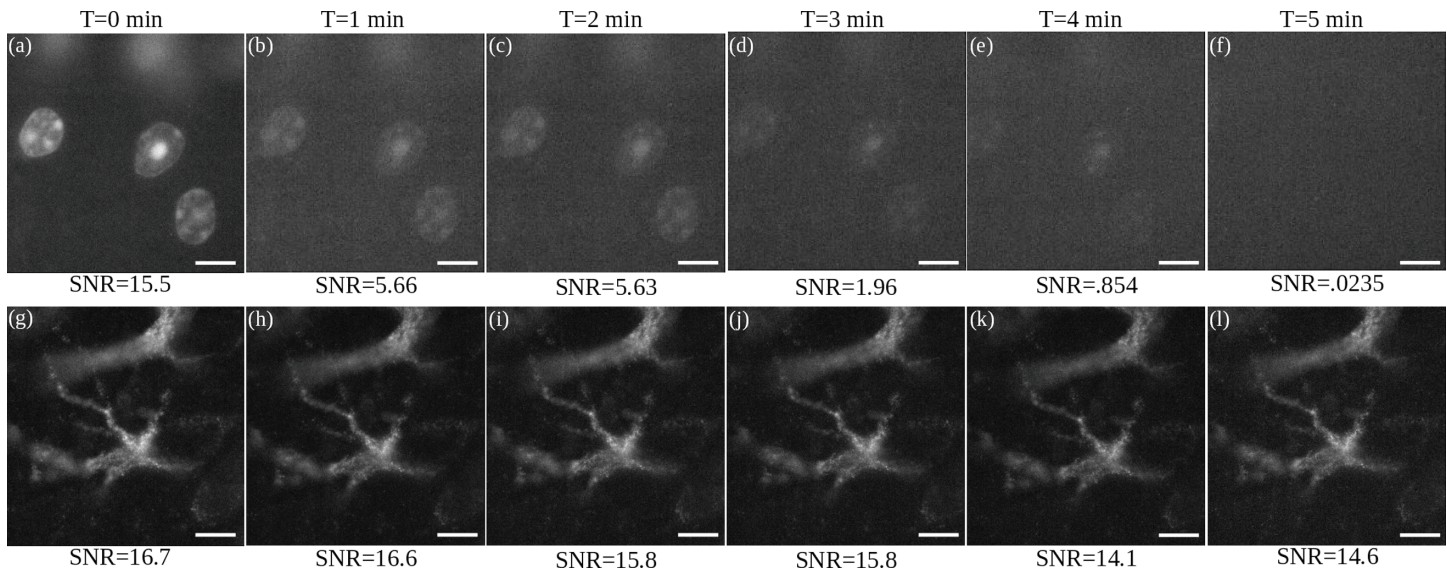

**Fig 3. Post-expansion photostability tests with Hoechst 34580 and QDots show complete loss of signal for Hoechst 34580, while minimal photobleaching is observed for QDots.** The 8.7% loss of intensity for QDots suggests that ExM protocols have some effect on their photostability. Timelapse measurements were taken for 5 min at 10 s intervals using a 40x water immersion objective (0.8NA). One-minute intervals were shown for Hoechst 34580 (a–f) and QDot 585 (g–l). Scale bar for the post-expansion images represents 20 $\mu$m (physical size) and 4.8 $\mu$m (structural size). The SNR (in dB) is given below each image.

## 3.6 Expansion microscopy deconvolution

Previously published research on ExM leveraged confocal imaging for 3D reconstruction [1–7,9]. Our experiments use wide-field microscopy combined with deconvolution to significantly reduce acquisition time. Fig 5 shows the maximum intensity projections of the triplex stained tissues before and after deconvolution using Huygens Software (Scientific Volume Imaging). Deconvolution improves contrast and resolution to produce sharper structures. In addition, the details with spatial frequencies outside the optical transfer function (OTF) due to the Poisson noise are suppressed by deconvolution. Compared with the raw images of astrocyte, myelin, and nuclei cells, deconvolution provides clear structural information and a simple and precise estimate of the cells. The finer details swamped by low-resolution out-of-focus light visualized with reduced contrast are reconstructed. The intensity histogram of Fig 5c & 5d, shown in Fig 5g for the Hoechst 34580 channel further demonstrates background and noise suppression. A post-deconvolution histogram is rightly skewed and the majority of the intensities fall to zero compared to the pre-deconvolution plot. Standard deviation values before and after deconvolution show that the intensities are tightly clustered around the mean after deconvolution, emphasizing the removal of out-of-focus signal. Comparison of line intensity profile before and after deconvolution Fig 5h shows the improved signal.

Deconvolution can create numerous systemic artifacts due to lamp flickering, data truncation, or ringing. Parameters were carefully optimized to minimize artifacts based on Huygens Software documentation for the classical maximum likelihood estimation (CMLE) algorithm. Furthermore, with an increasing number of iterations, resolution can also be slightly increased while preserving the structural information. Fig 6 compares confocal with widefield deconvolved images of an expanded brain tissue section. Deconvolution was performed using Huygens (Scientific Volume Imaging) version 20.10 with classic maximum

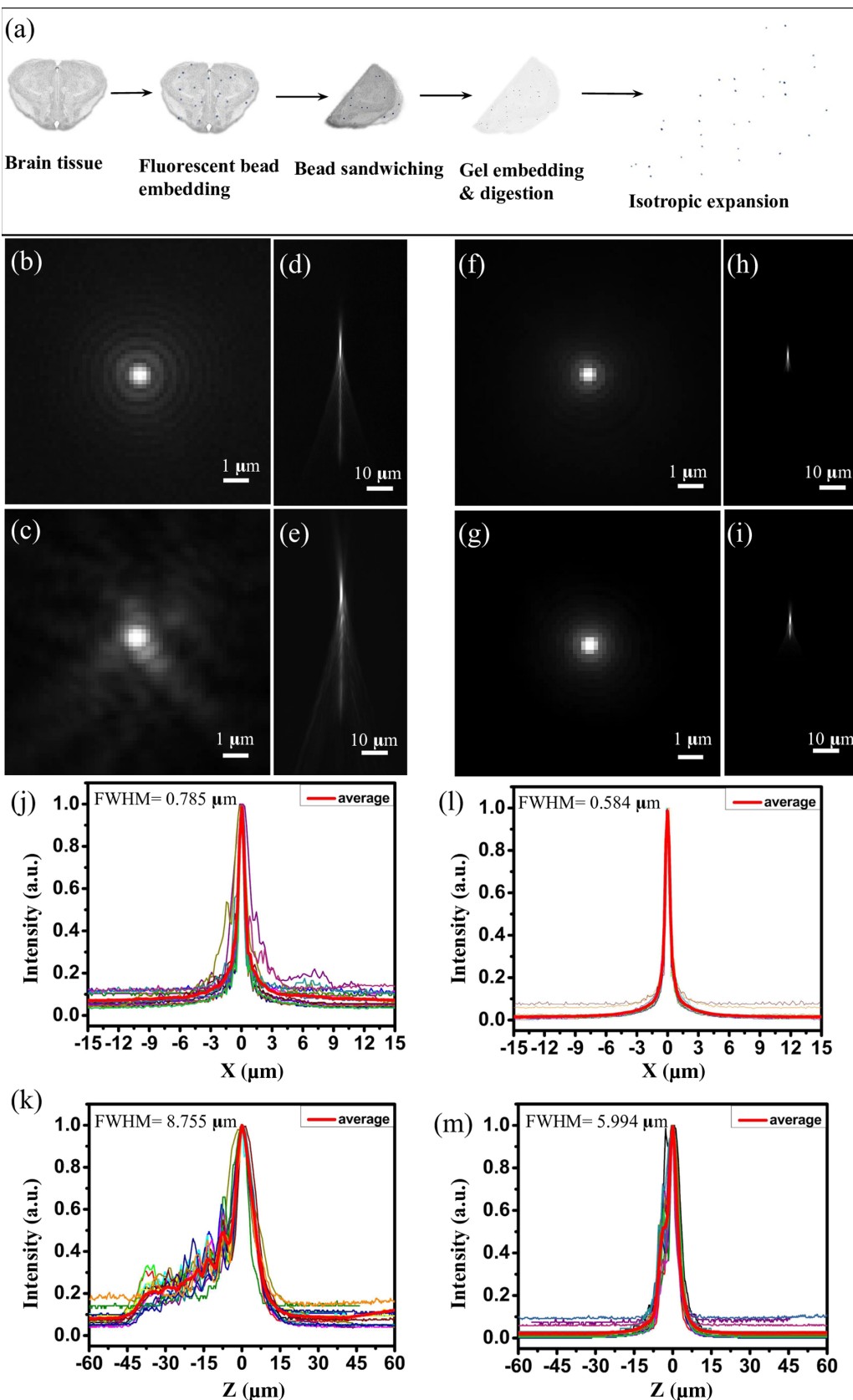

**Fig 4. Comparison of radial and axial point spread functions before and after expansion.** (a) Schematic illustration of the fluorescent phantom constructed by folding microbeads ($\approx 1\,\mu m$) within a 10 μm thick tissue section.

Image stacks were obtained with a 40x water dipping objective (0.8NA) at a physical interval of 0.8 μm with a wide-field fluorescence microscope. Pre-expansion profiles show a wide variability of scattering due to depth and tissue density: (b [lateral], d [axial]) minimal scattering and (c [lateral], e [axial]) significant scattering. Corresponding images after expansion are shown (f-i), demonstrating reduced scattering and a more consistent PSF. Average lateral and axial intensity profiles are shown for 15 different beads before (j, k) and after (l, m) expansion.

likelihood estimation (CMLE). Regularization parameters were tuned by adjusting the SNR between 30-40, and the background level was evaluated using the automated approach in the Deconvolution Wizard, with 94 iterations and a quality threshold of 0.02.

The x-z cross-section of the wide-field deconvolved image shows an axial elongation since the PSF changes along the optical axis. This is caused by the increased spherical aberration introduced by the sample thickness and mounting medium. In contrast, confocal PSF is equivalent to the square of the widefield PSF, since the light is passed through two apertures, significantly reducing out-of-focus light. The main advantage of widefield imaging is image acquisition time. Obtaining the 30 $\mu$m depth confocal image required 45 minutes, while the widefield acquisition of the same volume took approximately 6 minutes, providing 7.5 times increased acquisition speed. In contrast, spinning disk confocal microscopy (SDCM) aims for faster acquisition speeds within small fields of view to capture cell dynamics. This limits SDCM to small detectors and severely limits the number of photons acquired per pixel over time. Since ExM samples are not dynamic, widefield imaging can fully utilize high-resolution detector arrays.

The asymmetric behavior of the PSF can be reduced to some extent by modifying the refractive index of the immersion medium. In our case, the embedding medium was mostly water (ExM gel), and to match the refractive index values for the lens immersion used and the embedding medium, we used a 40x water immersion objective (0.8NA).

### 3.7 Pre-ExM Vs. Post-ExM

Fig 7 compares wide-field images of unexpanded and expanded deconvolved astrocytes. Structural features are more clearly visible after expansion. The star-shaped features with many extensions and end-feet encircling capillaries are clearly apparent in expanded astrocytes. Additional transparency induced by expansion greatly reduces scattering, providing a sharper post-deconvolution result. The x-z projections confirm the reduction of FWHM by ≈60% in the x-z plane with expansion. A Gaussian fit was used to determine the FWHM.

Aligned GFAP-labeled astrocytes are shown, comparing the results of both expansion microscopy and deconvolution (Fig 8). A broader example of the effects of deconvolution on both pre- and post-expanded samples is also shown (Fig 9). All images are acquired with a widefield microscope at 488 nm excitation with a lateral sampling rate of 0.16 μm/pixel, and an axial interval of 0.8 μm (note that ExM image stacks will be 4X - 5X larger to acquire the same z-axis range). Cerebral structural features are identified and used for image pre- and post-expansion registration. Random clusters are also observed, possibly due to the nonspecific binding of streptavidin [30]. It may be possible to reduce this clustering with optimized IHC protocols. Based on the Rayleigh criterion, the diffraction limit is ≈ 372 nm which is insufficient to capture the structural details in astrocytes. With expansion and deconvolution, sub-diffraction features (≈ 83 nm- 93 nm) are captured. This ultimately opens the door to

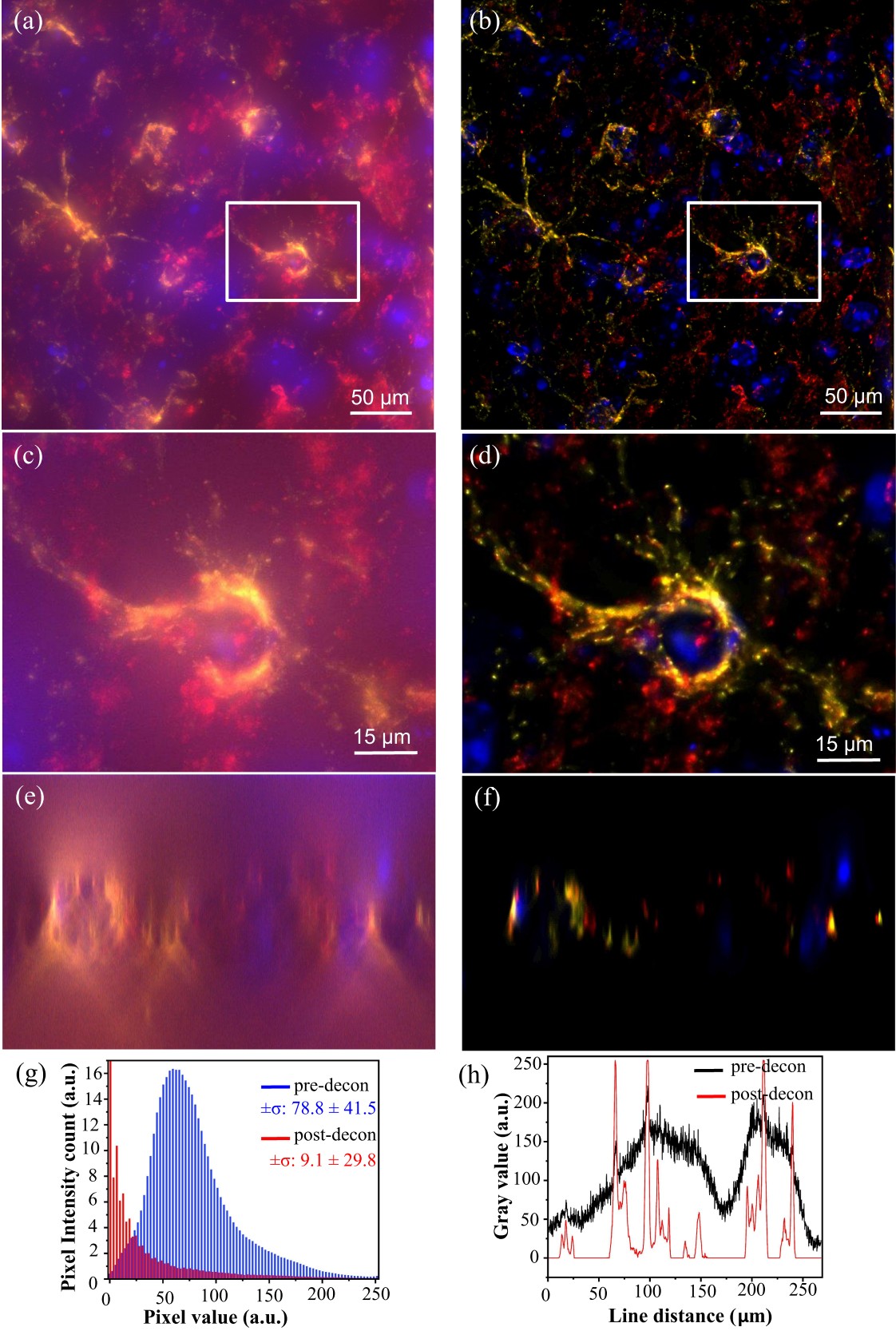

**Fig 5. Pre- and post-deconvolution images of the expanded brain tissue showing GFAP (QDot 585, yellow), MBP (QDot 655, red), nuclei (Hoechst 34580, blue).** A 3D image stack was obtained from the hippocampus at a physical interval of 0.8 μm using

a 40x water-immersion objective (0.8NA). Maximum intensity projections are shown for raw (a, c) and deconvolved (b, d) images. Close-up images (c, d) show an astrocyte cell body. X-Z cross-sections are also shown for raw (e) and deconvolved (f) images. The intensity profile of (a,b) for the Hoechst 34580 channel is plotted as a histogram in (g). To emphasize the pattern, the zeroth intensity value is compressed in the post-decon plot. The intensity of a line drawn across (a,b) for the Hoechst 34580 channel before and after deconvolution is shown in (h).

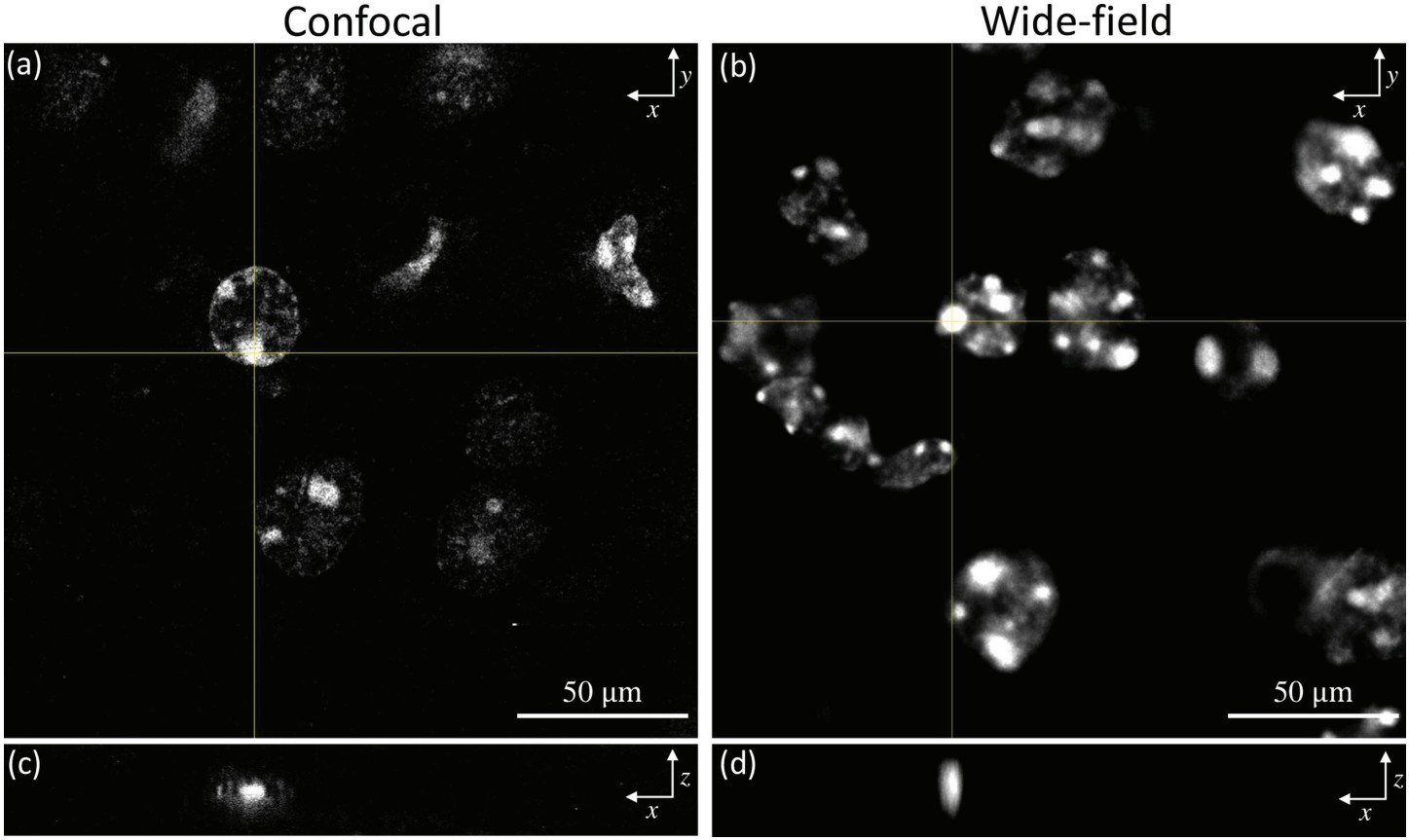

**Fig 6. Widefield deconvolved image comparisons of the expanded brain tissue with confocal microscopy image.** The tissue was first labeled with Hoechst-positive nuclei and then subjected to physical expansion. A 3D image stack was obtained at a physical interval of 0.6 μm and deconvolution was performed. A 40x objective (0.6NA) was used for both confocal and wide-field imaging. Single focal plane images are shown for the (a) confocal and (b) wide-field deconvolved images. An x-z intensity cross-section of a selected spot is also shown for (c) confocal and (d) wide-field deconvolved images.

quantitative analysis featuring the thickness/radius of an individual astrocyte. We further confirm the expansion factor of ≈ 4.1X by analyzing the astrocyte cell size before (15.46 μm) and after (63.81 μm) expansion.

## 4 Conclusion

In summary, our work demonstrates that integrating quantum dots (QDots) with expansion microscopy (ExM) offers a powerful solution to several challenges inherent to traditional confocal imaging. By combining the superior photostability and brightness of QDots with the

## Before-expansion & after-deconvolution

## After-expansion & after-deconvolution

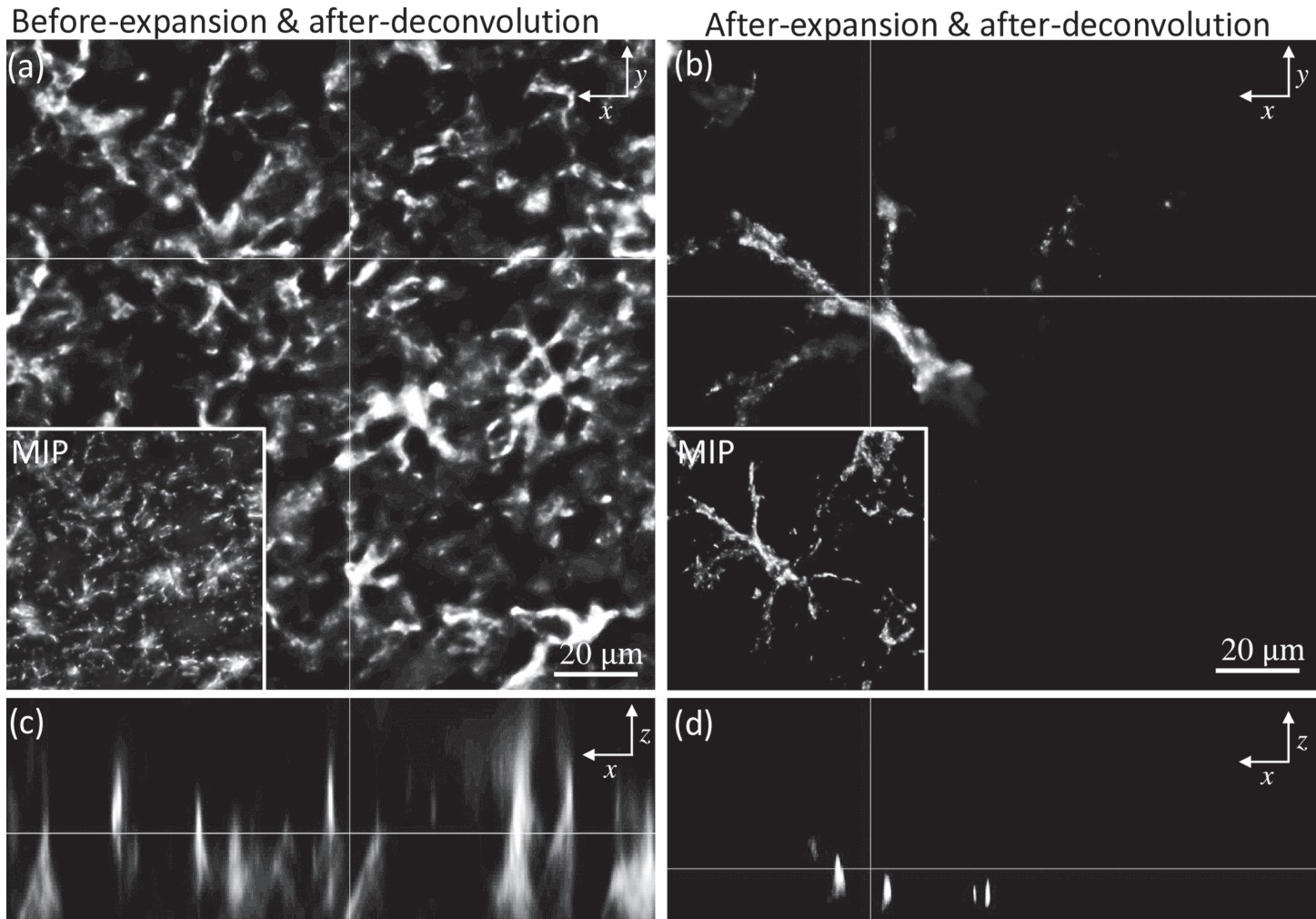

**Fig 7. Widefield image comparisons of the unexpanded brain tissue with expanded tissue after deconvolution.** The tissue was first labeled with GFAP-positive astrocytes and then subjected to physical expansion. 3D image stacks were obtained with a 40x water immersion objective (0.8NA). Single images are shown before (a) and after (b) expansion, with boxed insets showing maximum intensity projections of the entire stack. X-Z intensity cross-sections are also shown before (c) and after (d) expansion.

rapid acquisition capabilities of wide-field imaging and the resolution enhancement provided by deconvolution, we achieved significantly brighter images with reduced photobleaching. This approach not only addresses the limitations of conventional fluorescent dyes in ExM but also supports effective multiplexing through the use of QDots' narrow emission bandwidth of QDots and minimal crosstalk.

Despite these advantages, the use of QDots in immunolabeling presents challenges such as larger particle size, batch-to-batch variability, and a tendency to aggregate, which may require further protocol optimization or custom synthesis of conjugates [31]. In addition, fewer Qdot conjugates are commercially available, and in-house synthesis with a more elaborate optimization may be required for effective immunolabeling.

However, many technologies have emerged from the original ExM protocol that dispense with immunostaining [32]. Emerging techniques—such as click-ExM, which utilizes click-chemistry for the imaging of diverse biomolecules—provide promising alternatives that could

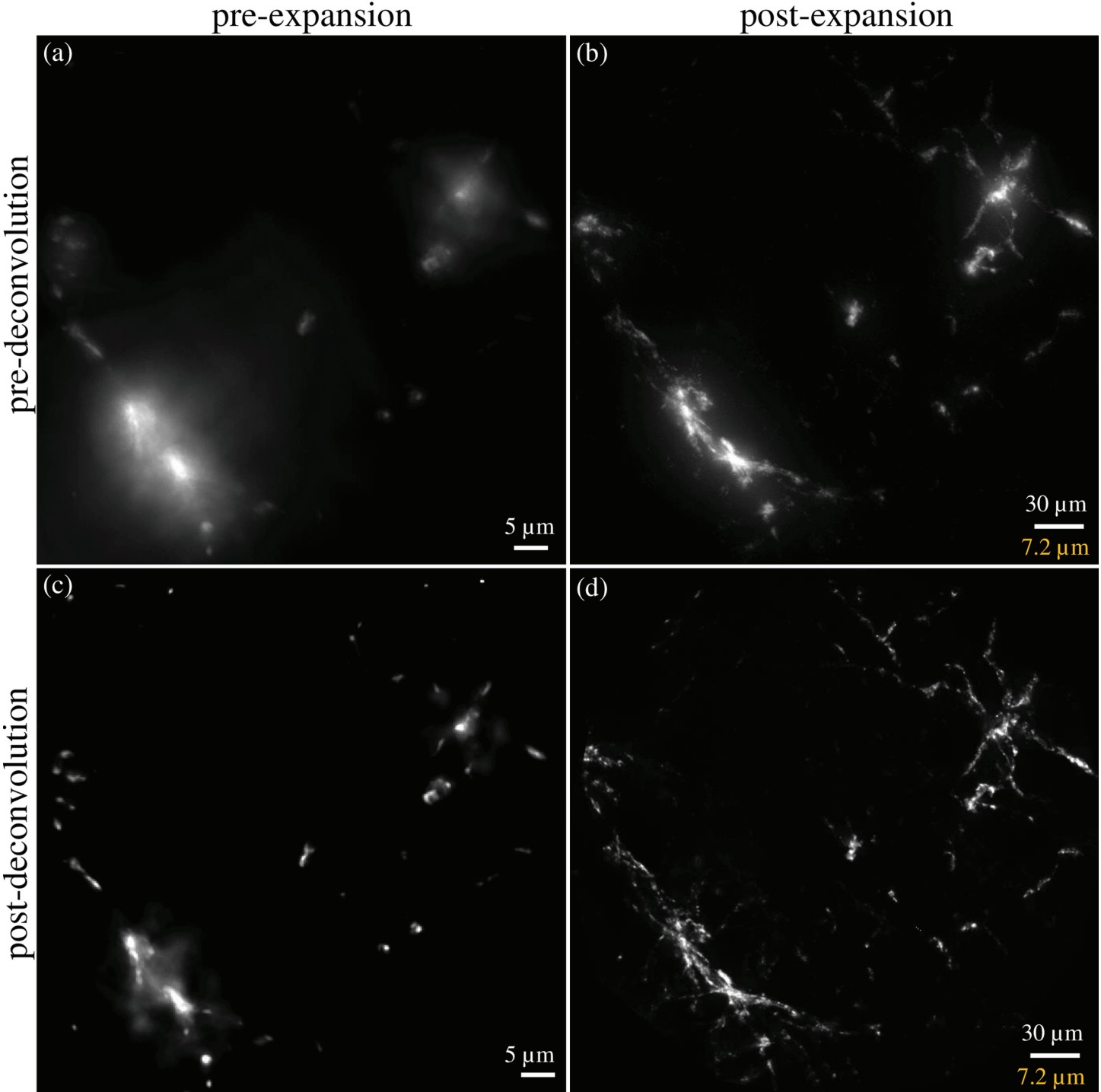

**Fig 8. Comparison of GFAP-labeled astrocytes both before and after expansion (a, b) and deconvolution (c, d).** Maximum intensity projections of wide-field image stacks in both native (a) and expanded (b) samples are shown. Both image stacks were obtained with a 40x water-immersion objective (0.8NA) at regular z-axis intervals of 0.8 μm. Maximum intensity projections of the deconvolved stacks are shown for both the native (c) and expanded cases. Expansion microscopy physically magnifies the sample ≈4.1X, and the scale bar for the post-expansion image represents the physical size (white) and structural size (yellow).

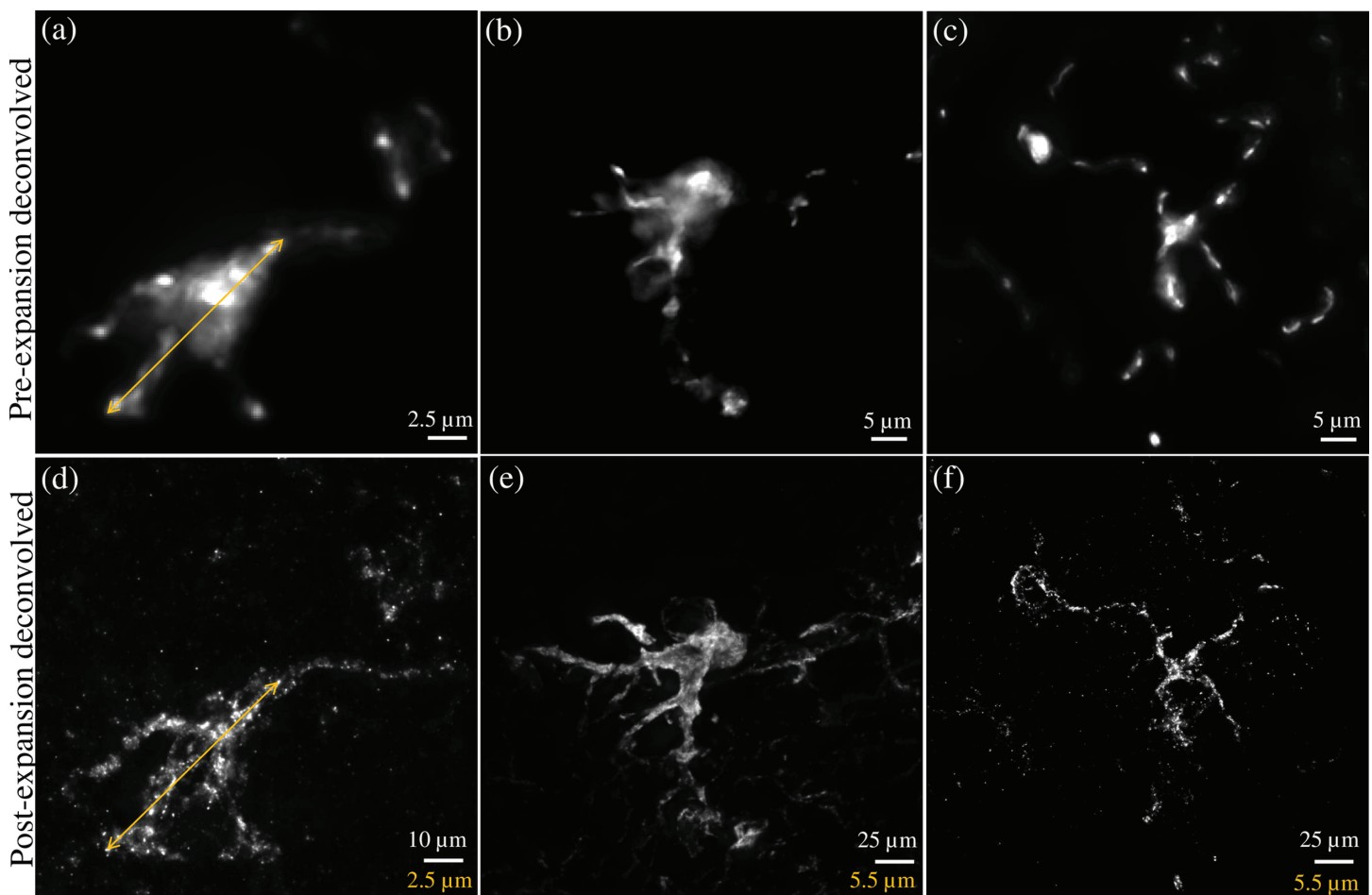

**Fig 9. Comparison of pre- (a–c) vs. post (d–f) expansion images of GFAP-positive astrocytes imaged with a widefield fluorescence microscope with a 40x water-immersion objective (0.8NA) and deconvolved with Huygens Software (Scientific Volume Imaging).** Expansion microscopy physically magnifies the sample ≈4.0–4.5X. The scale bar for the post-expansion image represents physical size (white) and structural size (yellow).

be combined with QDot labeling [33]. For example, replacing fluorescence-labeled streptavidin with QDot-conjugated streptavidin in these protocols could further enhance image quality and acquisition speed. Overall, our study establishes a foundation for high-resolution, time-efficient ExM imaging and highlights avenues for future refinement. These improvements promise to expand the applicability of ExM in detailed 3D imaging of complex biological tissues.

## Supporting information

**S1 Table. Photostability measurements.** Average intensity measured over time from Fig 2. The duration of excitation exposure is provided along with the average and normalized intensity of the Alexa dye and QDots.
(XLSX)

**S1 Fig. Photostability images.** Raw images acquired for Fig 2 showing photobleaching of the Alexa dye.
(ZIP)

**S2 Table. Point spread function data.** Table containing raw data used to generate the point spread function cross-sections in Fig 4.
(XLSX)

**S2 Fig. Point spread function data.** Raw cross-sections of the point spread function used to produce Fig 4.
(ZIP)

## Author contributions

**Conceptualization:** Loku Gunawardhana, Jiaming Guo, Camille Artur, Tasha Womack, Jason L. Eriksen, David Mayerich.

**Data curation:** Loku Gunawardhana.

**Formal analysis:** Loku Gunawardhana.

**Funding acquisition:** David Mayerich.

**Investigation:** Loku Gunawardhana, Jiaming Guo, Camille Artur.

**Methodology:** Loku Gunawardhana, Wilna Moree, Tasha Womack, David Mayerich.

**Resources:** David Mayerich.

**Software:** Loku Gunawardhana, David Mayerich.

**Supervision:** Jason L. Eriksen, David Mayerich.

**Validation:** Loku Gunawardhana, Jason L. Eriksen.

**Writing – original draft:** Loku Gunawardhana, Wilna Moree, Jason L. Eriksen, David Mayerich.

**Writing – review & editing:** Loku Gunawardhana, Wilna Moree, Jiaming Guo, Camille Artur, Tasha Womack, Jason L. Eriksen, David Mayerich.

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
