## [Decision Letter · Decision Letter 0]

PONE-D-24-54535Fast Photostable Expansion Microscopy Using QDots and DeconvolutionPLOS ONE

Dear Dr. Mayerich,

Thank you for submitting your manuscript to PLOS ONE. After careful consideration, we feel that it has merit but does not fully meet PLOS ONE’s publication criteria as it currently stands. Therefore, we invite you to submit a revised version of the manuscript that addresses the points raised during the review process.

 The manuscript presents the combination of Expansion Microscopy (ExM) with QDots labeling and wide-field image deconvolution, providing a high-resolution and time-efficient method for imaging biological tissues. The text is clear and well-structured, and the reviewers recommend it for publication once certain critical issues are addressed.

We look forward to receiving your revised manuscript.

Kind regards,

Luca Pesce, Ph.D.

Academic Editor

PLOS ONE

Journal Requirements:

[DM and JE are both stakeholders in SwiftFront, LLC and Wilna Moree holds a significant financial interest].

We note that one or more of the authors are employed by a commercial company: [SwiftFront, LLC]

Within your Competing Interests Statement, please confirm that this commercial affiliation does not alter your adherence to all PLOS ONE policies on sharing data and materials by including the following statement: ""This does not alter our adherence to PLOS ONE policies on sharing data and materials.” (as detailed online in our guide for authors http://journals.plos.org/plosone/s/competing-interests) . If this adherence statement is not accurate and there are restrictions on sharing of data and/or materials, please state these. Please note that we cannot proceed with consideration of your article until this information has been declared.

3. Thank you for stating the following in the Acknowledgments Section of your manuscript: [This work was funded in part by:

• (DM) National Institutes of Health / National Heart, Lung, and Blood Institute (NHLBI) #R01HL146745 (https://www.nhlbi.nih.gov/)

• (DM) National Science Foundation CAREER Award #1943455 285

(https://www.nsf.gov/) Sponsors did not play a role in the study design, data collection, analysis, decision to publish, or preparation of the manuscript. DM and JE are both stakeholders in SwiftFront, LLC and WM holds a significant financial interest.]

5. Your abstract cannot contain citations. Please only include citations in the body text of the manuscript, and ensure that they remain in ascending numerical order on first mention.

Reviewers' comments:

Reviewer's Responses to Questions

**Comments to the Author**

1. Is the manuscript technically sound, and do the data support the conclusions?

Reviewer #1: Yes

Reviewer #2: Yes

2. Has the statistical analysis been performed appropriately and rigorously? 

Reviewer #1: Yes

Reviewer #2: N/A

3. Have the authors made all data underlying the findings in their manuscript fully available?

Reviewer #1: Yes

Reviewer #2: Yes

4. Is the manuscript presented in an intelligible fashion and written in standard English?

Reviewer #1: Yes

Reviewer #2: Yes

5. Review Comments to the Author

Reviewer #1: The manuscript reports the combination of Expansion microscopy (ExM) with QDots labeling and deconvolution of wide-field images to offer a high-resolution, time-effective imaging approach of biological tissues. The text is well written and easy to follow, therefore I recommend it for publication, after some criticalities have been solved.

1. In general, I find figures to be too many and quite repetitive. I believe that a few figures can be combined together (e.g. Fig2 and 3), resulting in a more fluid reading of the paper.

2. The term “clearing” is not used properly in the text. “Tissue clearing” refers to active processes, whereas what you are referring to as a side effect is more an “increased optical transparency”.

3. Line 145: please change the term Alexa with “Alexa-Fluor”.

4. I believe it could be of interest for the final goal of the work to include a picture of a brain slice both before and after ExM, taken on graph paper, to confirm the rate of expansion that you measured at the microscale (4.1x).

5. In paragraph 3.4, could you please provide information on the laser intensity used, and if the measurements were performed at the same intensity?

6. Figure 4C is not clear to me. Why are the structures at the border brighter than in figure 4B? if this is due to modified contrasts, it is more correct to use the same settings in the two images. Moreover, the reference to panel C is missing in the caption.

7. Finally, the conclusion section does not add anything to the paper, except for summarizing the results. Please, discuss your findings and present further improvements that can be of interest. For example, in lines 239-243, you discuss how changing the refractive index can improve the outcome. It is also possible to match the refractive index of the expanded samples, and use a RI tunable lens for the imaging for better results.

Reviewer #2: The manuscript presents a novel approach to combining quantum dots (QDots) with expansion microscopy (ExM) to enhance imaging speed and photostability for super-resolution imaging. The study demonstrates the utility of QDots for multiplexed labeling and widefield imaging and validates their superiority over traditional fluorophores in terms of photostability and signal retention. The methodology and results are well-detailed and provide a significant advancement for researchers in microscopy and imaging sciences.

I really appreciate the novelty of the study and the comprehensive methodology. The use of QDots in ExM is a clear innovation that addresses challenges such as photobleaching and acquisition speed. The application of deconvolution further enhances the utility of the protocol.

Minor comments:

The introduction is well-written but assumes some familiarity with ExM. Adding a brief explanation of why ExM inherently reduces fluorescence signal and how QDots counteract this would enhance accessibility for readers unfamiliar with the technique.

I understand that the manuscript offers clear qualitative comparisons, such as photostability and resolution improvements. While a more quantitative evaluation of imaging resolution enhancement post-deconvolution (e.g., statistical analysis of FWHM across multiple samples) is not strictly necessary, it would significantly strengthen the claims.

The manuscript concludes with a strong emphasis on the utility of QDots for ExM but could benefit from a broader discussion of potential limitations (e.g., QDot aggregation or cost) and possible solutions.

6. PLOS authors have the option to publish the peer review history of their article (what does this mean?). If published, this will include your full peer review and any attached files.

Reviewer #1: No

Reviewer #2: No

---

## [Author Response · Author response to Decision Letter 1]

24 Feb 2025

The reviewers comments and advice was addressed and descriptions provided in the "Response to Reviewers" document.

---

## [Decision Letter · Decision Letter 1]

PONE-D-24-54535R1Fast Photostable Expansion Microscopy Using QDots and DeconvolutionPLOS ONE

Dear Dr. Mayerich,

Thank you for submitting your manuscript to PLOS ONE. After careful consideration, we feel that it has merit but does not fully meet PLOS ONE’s publication criteria as it currently stands. Therefore, we invite you to submit a revised version of the manuscript that addresses the points raised during the review process.

 The manuscript introduces a method integrating quantum dots, widefield imaging, and deconvolution to enhance photostability and speed in expansion microscopy. While promising, it lacks details on deconvolution parameters, photostability comparisons, and statistical validation of signal improvements. Addressing these gaps would strengthen the study before publication.

We look forward to receiving your revised manuscript.

Kind regards,

Luca Pesce, Ph.D.

Academic Editor

PLOS ONE

Journal Requirements:

Reviewers' comments:

Reviewer's Responses to Questions

**Comments to the Author**

1. If the authors have adequately addressed your comments raised in a previous round of review and you feel that this manuscript is now acceptable for publication, you may indicate that here to bypass the “Comments to the Author” section, enter your conflict of interest statement in the “Confidential to Editor” section, and submit your "Accept" recommendation.

Reviewer #1: All comments have been addressed

Reviewer #2: (No Response)

2. Is the manuscript technically sound, and do the data support the conclusions?

Reviewer #1: Yes

Reviewer #2: Yes

3. Has the statistical analysis been performed appropriately and rigorously? 

Reviewer #1: Yes

Reviewer #2: N/A

4. Have the authors made all data underlying the findings in their manuscript fully available?

Reviewer #1: Yes

Reviewer #2: Yes

5. Is the manuscript presented in an intelligible fashion and written in standard English?

Reviewer #1: Yes

Reviewer #2: Yes

6. Review Comments to the Author

Reviewer #1: I recommend this revised version of the paper for publication as the authors have adequately answered and revised the weaknesses found in the previous round of revision.

Reviewer #2: The manuscript presents a method that integrates quantum dots (QDots), widefield imaging, and deconvolution to improve photostability and acquisition speed in expansion microscopy (ExM). The study is relevant, as traditional ExM techniques suffer from signal dilution and photobleaching, which limit their practical use for large-scale, high-resolution imaging. The authors propose that combining QDots with widefield imaging and deconvolution overcomes these limitations, offering a more efficient approach.

Overall, the manuscript is well-structured and clearly written, with appropriate methodology and data presentation. However, some areas require further clarification and improvement with minor revision.

o Deconvolution Parameters: While the authors apply Huygens software for deconvolution, the exact PSF modeling parameters are missing.

o Photostability Comparisons: The time-lapse studies are valuable, but it would be useful to include a quantitative photostability index for QDots vs. Alexa dyes across multiple imaging sessions.

o The manuscript reports improvement in signal-to-noise ratio (SNR), but no statistical tests are provided to support this claim.

o A side-by-side quantitative comparison of SNR before and after deconvolution (e.g., mean intensity, contrast-to-noise ratio) should be included.

o In Fig. 7 (pre- and post-deconvolution images), an objective numerical comparison of resolution enhancement (e.g., FWHM of PSFs before and after processing) should be provided.

Recommendation: the study presents a novel application of QDots in ExM, but methodological gaps, data quantification, and statistical analysis should be improved before publication. Addressing the quantitative comparisons, SNR/statistical validation, and deconvolution parameters would strengthen the manuscript.

7. PLOS authors have the option to publish the peer review history of their article (what does this mean?). If published, this will include your full peer review and any attached files.

Reviewer #1: No

Reviewer #2: No

---

## [Decision Letter · Decision Letter 2]

Fast Photostable Expansion Microscopy Using QDots and Deconvolution

PONE-D-24-54535R2

Dear Dr. Mayerich,

We’re pleased to inform you that your manuscript has been judged scientifically suitable for publication and will be formally accepted for publication once it meets all outstanding technical requirements.

Kind regards,

Luca Pesce, Ph.D.

Academic Editor

PLOS ONE

Additional Editor Comments (optional):

Reviewers' comments:

Reviewer's Responses to Questions

**Comments to the Author**

1. If the authors have adequately addressed your comments raised in a previous round of review and you feel that this manuscript is now acceptable for publication, you may indicate that here to bypass the “Comments to the Author” section, enter your conflict of interest statement in the “Confidential to Editor” section, and submit your "Accept" recommendation.

Reviewer #2: All comments have been addressed

2. Is the manuscript technically sound, and do the data support the conclusions?

Reviewer #2: Yes

3. Has the statistical analysis been performed appropriately and rigorously? 

Reviewer #2: Yes

4. Have the authors made all data underlying the findings in their manuscript fully available?

Reviewer #2: Yes

5. Is the manuscript presented in an intelligible fashion and written in standard English?

Reviewer #2: Yes

6. Review Comments to the Author

Reviewer #2: I recommend this revised version of the paper for publication as the authors have

adequately answered and revised the weaknesses.

7. PLOS authors have the option to publish the peer review history of their article (what does this mean?). If published, this will include your full peer review and any attached files.

Reviewer #2: No

---

## [Editor Report · Acceptance letter]

PONE-D-24-54535R2

PLOS ONE

Dear Dr. Mayerich,

I'm pleased to inform you that your manuscript has been deemed suitable for publication in PLOS ONE. Congratulations! Your manuscript is now being handed over to our production team.

Kind regards,

on behalf of

Dr. Luca Pesce

Academic Editor

PLOS ONE